# Functionalized Titanium Dioxide Nanoparticle-Based Electrochemical Immunosensor for Detection of SARS-CoV-2 Antibody

**DOI:** 10.3390/diagnostics12112612

**Published:** 2022-10-27

**Authors:** Mohd Abubakar Sadique, Shalu Yadav, Vedika Khare, Raju Khan, Gagan Kant Tripathi, Purnima Swarup Khare

**Affiliations:** 1Industrial Waste Utilization, Nano and Biomaterials, CSIR-Advanced Materials and Processes Research Institute (AMPRI), Bhopal 462026, India; 2Academy of Scientific and Innovative Research (AcSIR), Ghaziabad 201002, India; 3School of Nanotechnology, Rajiv Gandhi Proudyogiki Vishwavidyalaya, Bhopal 462033, India

**Keywords:** titanium dioxide, chitosan, SARS-CoV-2 N protein, SARS-CoV-2 antibody, electrochemical immunosensor

## Abstract

The advancement in biosensors can overcome the challenges faced by conventional diagnostic techniques for the detection of the highly infectious severe acute respiratory syndrome coronavirus 2 (SARS-CoV-2). Hence, the development of an accurate, rapid, sensitive, and selective diagnostic technique can mitigate adverse health conditions caused by SARS-CoV-2. This work proposes the development of an electrochemical immunosensor based on bio-nanocomposites for the sensitive detection of SARS-CoV-2 antibodies through the differential pulse voltammetry (DPV) electroanalytical method. The facile synthesis of chitosan-functionalized titanium dioxide nanoparticles (TiO_2_-CS bio-nanocomposites) is performed using the sol-gel method. Characterization of the TiO_2_-CS bio-nanocomposite is accomplished using UV-vis spectroscopy, Raman spectroscopy, X-ray diffraction (XRD), and transmission electron microscopy (TEM). The electrochemical performance is studied using cyclic voltammetry (CV), DPV, and electrochemical impedance spectroscopy (EIS) for its electroanalytical and biosensing capabilities. The developed immunosensing platform has a high sensitivity with a wide range of detection from 50 ag mL^−1^ to 1 ng mL^−1^. The detection limit of the SARS-CoV-2 antibody in buffer media is obtained to be 3.42 ag mL^−1^ and the limit of quantitation (LOQ) to be 10.38 ag mL^−1^. The electrochemical immunosensor has high selectivity in different interfering analytes and is stable for 10 days. The results suggest that the developed electrochemical immunosensor can be applicable for real sample analysis and further high-throughput testing.

## 1. Introduction

The coronavirus 2019 (COVID-19) epidemic ultimately resulted in an unmanageable condition, which led to its designation as a pandemic on a worldwide scale. The real-time polymerase chain reaction (RT-PCR) is the standard, most dependable way of diagnosing COVID-19 [1]. Nevertheless, it calls for the shipment of samples to highly developed labs and necessitates a considerable amount of time to amplify the viral genome [2,3]. As a result, there is an immediate need for a detection kit that is portable, accurate, and quick. Biosensors have huge potential due to their low cost and the fact that they respond to real-time analysis. Detection technology that is based on nanoparticles (NPs) has recently been developed, and tests have shown that it is superior to traditional approaches in terms of its selectivity and sensitivity [4,5]. Nanomaterials are distinguished from bulk materials by their exceptional physical, chemical, electrical, and optical capabilities to combat viral infections [6,7]. Nanomaterials operate on the same scale as biological processes and are easily functionalized with substrates of interest offering significant utilization in the development of biosensors [8,9]. Nanomaterials, such as metal NPs, metal oxides, transition metal dichalcogenides, quantum dots, and carbon-based nanomaterials can improve the accuracy and sensitivity of biosensors, allowing them to produce reproducible and consistent results [10,11,12].

Over the past years, numerous biosensing approaches for various infectious viruses have been reported, such as surface plasmonic resonance [13], fluorescence [14], and surface-enhanced Raman spectroscopy (SERS) [15], as well as colorimetric [16,17], electrical [18,19,20,21], lateral flow assay [22,23], and electrochemical methods [24,25,26]. Out of these, electrochemical techniques exhibit numerous advantages such as cost-effectiveness, ease of use, portability, highly specific and sensitive results, accuracy, and miniaturization ability [24,27,28]. In the field of medical diagnostics, electrochemical biosensors have proven to be effective in the detection of different viruses [29,30,31,32]. Several studies have been reported for electrochemical-based detection of the SARS-CoV-2 virus; for instance, Sadique et al. reported a graphene oxide–gold nanocomposite-based electrochemical method for the dual detection of SARS-CoV-2 antigen and antibody with high sensitivity [33]. In another study, a polydopamine-decorated molybdenum disulfide nanosheet-based electrochemical platform was developed for the highly selective and sensitive detection of SARS-CoV-2 nucleocapsid (N) protein [34].

For the development of highly sensitive electrochemical immunosensors, smart nanomaterials such as tin oxide (SnO_2_) [35], titanium dioxide (TiO_2_) [36], graphene oxide [37], zirconium oxide (ZrO_2_) [38], zinc oxide (ZnO) [39], cerium oxide (CeO_2_) [40], iron oxide (Fe_3_O_4_) [41,42], and gold NPs (AuNPs) [43] have been explored. In addition, nano-structuring the electrode surface can further improve the performance of the electrochemical immunosensor. The enhanced electrochemical reaction rate generated by the expanded electrode surface area-to-volume ratio due to the presence of nanostructures on the electrode [44]. In the research conducted by Chin et al., carbon NPs modified electrodes improved electron charge transfer kinetics and raised the current feedback by 63% for the detection of the encephalitis virus [45].

Many nanoparticles (NPs) have been recently employed for the detection of SARS-CoV-2 antibodies using different electrochemical techniques. For instance, Li et al. developed high-performance EIS μPADs nanobiosensors.that were capable of differentiating the concentrations of IgG antibody (CR3022) for SARS-CoV-2 in human serum samples, demonstrating the efficacy of these devices for COVID-19 diagnosis [46]. Alternatively, Liv et al. fabricated an electrode platform based on gold nanoparticles clusters, mercaptoethanol, the spike protein of the SARS-CoV-2 antigen, and a bovine serum albumin-modified glassy carbon electrode capable of detecting the SARS-CoV-2 spike antibody. The biosensing platform can identify 0.03 fg mL^−1^ of the SARS-CoV-2 spike antibody in samples of spiked saliva or oropharyngeal swabs and synthetic media in 35 min. The SARS-CoV-2 spike antibody is therefore linearly responded to by the technique from 0.1 fg mL^−1^ to 10 pg mL^−1^ [47]. Rahmati and the group used nickel hydroxide NPs (Ni(OH)_2_ NPs) for the voltammetric detection of the SARS-CoV-2 antibody with high sensitivity and a LOD of 0.3 fg mL^−1^ [48]. Lorentz et al. developed a Poly (3,4-ethylenedioxythiophene)–gold nanoparticles (PEDOT/AuNPs)-based impedimetric immunosensor for the detection of SARS-CoV-2 antibodies. The immunosensor exhibit a linear and rapid response of 30 min [49]. A study was carried out by Cardoso et al. to demonstrate a sensitive electrochemical biosensor that can detect minute levels of antibodies in human serum, despite being manufactured using a remarkably easy process. It had a detection limit of 0.7 pg mL^−1^ and had a linear response between 1.0 pg mL^−1^ and 10 ng mL^−1^ [50]. Furthermore, Durmus et al. reported iron oxides as magnetic NPs for real sample diagnosis of the SARS-CoV-2 virus [51]. Several electrochemical biosensors for SARS-CoV-2 detection have been listed in Table 1.

Due to their biocompatibility, extensive surface area, and exceptional adsorptive ability on a variety of electrode materials, TiO_2_ NPs are employed in the development of an electrochemical immunosensor. The ability of TiO_2_ NPs to aggregate is one of their main disadvantages in the fabrication of electrochemical immunosensors. It has been shown that, when nano-TiO_2_ interacts with biopolymers (starch, gums, and chitosan (CS)), the spontaneous agglomeration of TiO_2_ is reduced, improving the functional qualities of composites [52,53].

CS is a biopolymer made from chitin (a polysaccharide constituent of crustacean shells). Owing to its innocuous, exceptional film-forming ability, outstanding adherence, and impulsive toughness, its use as a bioactive material in sensor applications has increased. The amino groups present in CS serve as ligands and combine with TiO_2_ NPs to produce a CS-Ti complex that has greater stability [54,55]. The addition of CS to TiO_2_ NPs improves its conductivity as well as the sensitivity, stability, and biomolecular-sensing capabilities of the immunosensor due to the high electrocatalytic performance of CS [56].

In this study, we have synthesized and evaluated the capability of chitosan-functionalized TiO_2_ NPs (TiO_2_-CS) bio-nanocomposites for the electrochemical detection of the SARS-CoV-2 antibody. TiO_2_ is synthesized by a simple, eco-friendly, quick, and economical approach, and CS functionalization is achieved through the sol-gel method. Subsequently, the prepared TiO_2_-CS bio-nanocomposite is efficiently utilized in the fabrication of an electrochemical immunosensor to detect SARS-CoV-2 antibodies within 5 min. The electrochemical immunosensor exhibits high sensitivity and selectivity in a wide detection range between 50 × 10^−18^ g mL^−1^ (ag mL^−1^) and 1.0 × 10^−9^ g mL^−1^ (ng mL^−1^) with a lower detection limit of 3.42 ag mL^−1^. The results suggest that the immunosensing platform is a promising candidate for clinical applications.

## 2. Materials and Methods

### 2.1. Materials and Reagents

Titanium (IV) isopropoxide (TTIP, 97%), chitosan (CS, low molecular weight), potassium ferricyanide (III) [K_3_Fe(CN)_6_], potassium hexacyanoferrate trihydrate [K_4_Fe(CN)_6_·3H_2_O], N-(3-(dimethylamino)propyl)-N’-ethylcarbodiimide hydrochloride (EDC), N-hydroxysulfosuccinimide sodium salt (NHS), and bovine serum albumin (BSA) were acquired from Sigma-Aldrich, Burlington, MA, USA. Potassium chloride (KCl), sodium phosphate monobasic dehydrate (NaH_2_PO_4_·2H_2_O), sodium phosphate dibasic (Na_2_HPO_4_), and sodium hydroxide (NaOH) were bought from Merck, India. Glacial acetic acid (CH₃COOH), sulfuric acid (H_2_SO_4_), orthophosphoric acid (H_3_PO_4_), and hydrochloric acid (HCl) were procured from RANKEM. SARS-CoV-2 nucleocapsid protein (ab273530), and anti-nucleocapsid SARS-CoV-2 immunoglobulin (Ig) G (ab273167) were procured from Abcam, UK. The rest of the solvents and chemicals were of analytical grade and used without further refining. All the solutions were prepared by ultrapure milli-Q water (18.3 MΩ) generated from a Millipore instrument.

### 2.2. Synthesis of Titanium Dioxide (TiO_2_)

TiO_2_ NPs were prepared by the previously reported method with some modifications [57]. The detailed method is mentioned in the Appendix A.

### 2.3. Synthesis of Chitosan-Functionalized Titanium Dioxide (TiO_2_-CS) Bio-Nanocomposite

For the preparation of the bio-nanocomposite, 150 mg chitosan was dissolved in 30 mL of 0.05 M acetate buffer (pH 4.2) at ambient temperature and continuously stirred for 4 h. After 15 min of sonication, the CS solution was filtered to eliminate any undissolved CS. The previously synthesized TiO_2_ NPs were stirred for 30 min to disseminate them in the CS solution. Finally, a viscous TiO_2_–CS bio-nanocomposite was obtained. To form a 0.5 mg mL^−1^ solution, deionized water was utilized to dilute the viscous TiO_2_–CS solution. The above solution was kept under refrigeration until further use. Figure 1a represents the steps involved in the synthesis of TiO_2_ nanoparticles and TiO_2_-CS bio-nanocomposite.

### 2.4. Pre-Treatment of the Working Electrode

For the working electrode, the glassy carbon electrode (GCE) was used further for modifications. The electrode has a diameter of 0.3 cm (geometrical surface area 0.070 cm^2^). The electrode was bath-sonicated with 0.1 M of H_2_SO_4_ for 5 min and then washed with ethanol and water alternatively, for 2 min each. Then, the electrode was cleaned by hand polishing it on a microfiber cloth containing 0.3 μm and 0.05 μm grains of Al_2_O_3_ slurry. After this, the electrode was sonicated with milli-Q water for 2 min to remove the alumina particles from the GCE surface and kept in a vacuum desiccator before further use.

### 2.5. Assembly of the Immunosensing Platform for SARS-CoV-2 Antibody Detection

To fabricate the bio-nanocomposite-modified electrode, the GCE was modified by 5 µL of TiO_2_-CS (0.5 mg mL^−1^) and drop-casted on the surface of the electrode to create the bio-nanocomposite layer. For further characterization and modifications, the surface-modified GCE was dried for 23 h at ambient temperature. Then, the TiO_2_-CS bio-nanocomposite-altered GCE electrode was coated with 5 µL of EDC/NHS in a 4:1 ratio for 2 h to activate carboxylic groups at the termini [58]. The electrode was washed with phosphate-buffered saline (PBS, pH 7.0) to eliminate the unbound EDC/NHS. Then, 5 µL of SARS-CoV-2 antigen (5µg mL^−1^) (in PBS, pH 7.4) was drop-cast onto the surface-activated electrode (GCE/TiO_2_-CS/Antigen) and incubated overnight at 4 °C for the formation of the electrochemical immunosensor. The electrode was then treated with PBS (pH 7.0) to eliminate any remaining unbinding proteins [49]. Finally, the modified electrode was treated with 5 µL of 0.1% BSA in PBS (pH 7.0) and nurtured for 1 h to impede unbounded sites on the electrode surface (GCE/TiO_2_-CS/Antigen/BSA). The electrode was then treated with PBS (pH 7.0) to remove excess BSA. To validate the successful surface modification of GCE, the electrode was electrochemically characterized after each step. The immunosensing platform for the SARS-CoV-2 antibody was finally ready for detection experiments [33]. Figure 1b shows the fabrication steps for the development of the immunosensing platform for the SARS-CoV-2 antibody.

### 2.6. Morphological Analysis and Electrochemical Measurements

The synthesized TiO_2_, TiO_2_-CS bio-nanocomposite, and altered electrodes were characterized for their structural, morphological, optical, and electrochemical properties using UV-vis spectroscopy (Evolution 220, Thermo Fisher Scientific, Waltham, MA, USA), Raman spectroscopy (IndiRam CTR-300, Technos instruments, Rajasthan, India), X-ray diffraction (Rigaku, Miniflex-1), transmission electron microscopy (JEOL, JEM-F200), and electrochemical analysis (Autolab PGSTAT204, P/G Metrohm equipped with NOVA 2.1.4 software), respectively.

The subsequent alteration of electrodes and fabrication of the electrochemical immunosensor was studied through electrochemical experiments. The three electrodes system was set up as follows: Ag/AgCl (3.0 mol L^−1^) as the reference electrode, glassy carbon (GC) as the working electrode, and a platinum wire as the auxiliary electrode. A measure of 0.1 M phosphate-buffered saline (PBS) (pH 7.4) with 5 mM of ferri/ferrocyanide and 0.1 M KCl was used as the redox solution for the electrochemical measurements. Cyclic voltammetry (CV) and differential pulse voltammetry (DPV) techniques were used to conduct the electrochemical study with an appertained potential range from −0.3 to +0.8 V with 20 mV/s as the scan rate. The electrochemical impedance spectroscopy (EIS) method was also employed for the study, with a frequency range of 0.1–1 MHz.

### 2.7. Sample Preparation

The detection samples were made at various dilutions from a stock solution of 5 µg mL^−1^ SARS-CoV-2 antibody in PBS (pH 7.2). In redox solution, dilutions ranging from 1 ng mL^−1^ to 50 ag mL^−1^ were prepared. Before further usage, the samples were stored at −20 °C.

## 3. Results and Discussion

To verify and validate the study, the prepared bio-nanocomposite and electrochemical immunosensor were analyzed using various methods. The bio-nanocomposite was analyzed by UV-vis spectroscopy, Raman, XRD, and TEM methods. In addition, the electrochemical immunosensor was characterized using CV, DPV, and EIS electrochemical techniques.

### 3.1. Optical Studies

To investigate the optical response, UV-visible absorption spectra for the synthesized nanomaterials were obtained as shown in Figure 1a. TiO_2_ NPs show an absorption peak at 229 nm. In the tetrahedral symmetry, this absorption band is indicative of O_2p_ → Ti_3d_ transitions [59]. TiO_2_-CS bio-nanocomposite has an absorption peak of nearly 257 nm. The absorbance peak was red-shifted in the bio-nanocomposite, which may be ascribed to the surface alteration of TiO_2_ NPs generated by the chemical adsorption of CS on the TiO_2_ NPs surface [60].

The vibrational properties of the TiO_2_ NPs and TiO_2_-CS bio-nanocomposite were further analyzed by Raman scattering, as displayed in Figure 1b with typical vibrational bands of the TiO_2_ NPs and TiO_2_-CS bio-nanocomposite [61]. The Raman shifts of 146, 398, 514, and 640 cm^−1^ corresponded to the anatase phase of TiO_2_ NPs. There had been a decrease in peak intensities when TiO_2_ NPs and TiO_2_-CS were evaluated. When compared to TiO_2_ NPs, the majority of anatase TiO_2_ vibration modes in TiO_2_-CS bio-nanocomposite showed low intensities on the same wavenumber which ascribed to the successful loading of CS at TiO_2_ NPs [62]. Hence, the successful synthesis of TiO_2_-CS bio-nanocomposite is supported by the characterization results.

### 3.2. Structural Analysis

The X-ray diffraction patterns of synthesized TiO_2_ NPs and TiO_2_-CS bio-nanocomposite samples were depicted in Figure 2. The diffraction peaks of pure TiO_2_ NPs are studied at 2θ values of 25.28°, 38.7°, 48.05°, 53.89°, 55.06°, 62.69°, 68.74°, 70.16°, and 75.14° which correlate to the tetragonal anatase crystalline phases, and their corresponding *hkl* planes were (101), (004), (200), (105), (211), (204), (116), (220), and (215), respectively. The whole TiO_2_ diffraction pattern was associated with JCPDS No 21–1272 [63]. Using Bragg’s equation, we were able to determine that the appropriate d-spacing value for the maximum intensity plane (101) is 0.350 nm. This plane displays strong intensity at 25.28° in comparison with all of the other peaks. Consequently, the diffraction peaks for the prepared TiO_2_-CS bio-nanocomposite sample were found at the 2θ values of 26.92° and 36.06° with some small peaks at 44.38°, 53.72°, 56.36°, 60.00°, 66.00°, 68.44°, and 74.84°. All 2θ values are connected with the anatase phase of TiO_2_. The tetragonal configuration of TiO_2_ is unaffected by the presence of chitosan. Hence, the XRD data showed that the TiO_2_-CS bio-nanocomposite was successfully synthesized. In a thorough analysis of the XRD results, the presence of chitosan in the bio-nanocomposite system reduces the intensity of the diffraction peaks as well as the broadness of the peak. The average crystal size of the TiO_2_ NPs was calculated to be ~9.75 nm by the Scherrer formula (Equation (1)).
(1)$$D=\frac{K.\lambda }{\beta .\;Cos\;\left(\theta \right)}$$
where *D* represents the crystallite size (nm), *K* is the shape factor, *λ* is for X-ray wavelength (Å), *β* represents the full-width half maximum of the diffraction peak (radian), and θ is for Bragg angle (degree).

### 3.3. Morphological Investigation

The TEM investigation was performed to confirm the size of the particle, orientational growth, and particle distribution [64]. It is discovered that, at different magnifications (Figure 3), the surfaces of TiO_2_ NPs had a spherical crystal. At a scale of 100 nm, it was discovered that the aggregated particles of the produced TiO_2_ NPs had the form of spheres (Figure 3a). The particle distribution graph of TiO_2_ NPs is shown in the inset of Figure 3a, where it is shown that the average particle size was 11.5 nm. The high-resolution (HR) TEM picture shown in Figure 3b validates the d-spacing calculation that was pre-calculated from the XRD data. The image shows that the interatomic spacing of TiO_2_ NPs is around 0.31 nm. Figure 3c illustrates the TiO_2_ NPs selected area electron diffraction (SAED) pattern. This pattern demonstrated the crystalline nature of TiO_2_ NPs and crystal planes, which justified the XRD data. The presence of concentric rings that contain clear spots demonstrates the crystalline nature of TiO_2_ NPs. As shown in Figure 3d,e, the production of TiO_2_ was validated by the presence of a homogeneous distribution of TiO_2_ NPs embedded in the CS layer with HR-TEM image of TiO_2_-CS bio-nanocomposite, respectively. It seems that the TiO_2_-CS bio-nanocomposite (Figure 3f) is semicrystalline, since the diffused concentric rings, in addition to dispersed spots, are present.

### 3.4. Electrochemical Analysis

Before detecting the SARS-CoV-2 antibody, the electrochemical activity of the prepared bio-nanocomposite, surface-altered GCE, and designed electrochemical immunosensor was thoroughly investigated. CV, *DPV*, and EIS were first used to characterize the electrochemical immunosensor. Every electrochemical measurement was performed in a redox solution. CV was utilized to depict voltammetric profiles on electrode surfaces with well-defined oxidation and reduction peaks at each functionalization step.

As shown in Figure 4a, the electrochemical characterization carried out by CV tracks the individual steps of the electrochemical immunosensor assembly process. The CV of the bare GCE revealed unambiguous reduction and oxidation peaks due to the Fe^3+^/Fe^2+^ redox couple. The GCE/TiO_2_ electrode shows a rise in peak current values with a shift in peak potentials, indicating that the TiO_2_-altered electrode surface includes more conducting species. The TiO_2_-CS-altered surface (GCE/TiO_2_-CS) demonstrated a considerable increment in the peak current and change in the potential, which also indicate that the electrochemical characteristics of TiO_2_ NPs can be enhanced by dispersing in an electro-active CS biopolymer. As the electrochemical immunosensor assembly progressed, the inclusion of EDC/NHS stimulated cross-linking groups with the stages of antigen immobilization, resulting in a reduction in the anodic and cathodic peak currents of the redox pair. This happens due to the habitation of insulating protein molecules in the electrode surface (GCE/TiO_2_-CS/Antigen) that hinder electron transport at the electrode-electrolyte interface. Furthermore, BSA occluded the boundless electrode surface, preventing the ferri/ferrocyanide redox pair from diffusing, as demonstrated by the low peak current and a potential shift in the CV curve. This indicates that the electrochemical immunosensor (GCE/TiO_2_-CS/Antigen/BSA) was successfully constructed. The modified electrodes demonstrate the great sensitivity of the DPV method, with the TiO_2_-CS bio-nanocomposite having the highest peak current and the electrochemical immunosensor having the lower peak current, nearly as low as that of the bare GCE after BSA blocking, with some significant potential shifts. Figure 4b shows DPV curves that are compatible with the CV findings stated above in Figure 4a.

Afterward, the Randles–Sevcik equation was adopted to enumerate the effective surface area (A) of the electrode using scan rate (*ν*) and peak current (Ip) in an observed voltammogram. It is expressed in Equation (2) as:(2)$$I_{p}=(2.687\times 10^{5})\;n^{3/2}\;{\mathrm{AD}}^{1/2}\;Cv^{1/2}$$

When it comes to redox reaction cycles, such as the one involving ferrocene and ferrocenium, the peak current (Ip) is dictated by the concentration of redox-active species (C) which is 5 × 10^−6^ mol cm^−3^. The diffusion coefficient of those species (D = 7.26 × 10^−6^ cm^2^ s^−1^), the scan rate (*v* = 20 mV s^−1^), the number of electrons transferred in the redox reaction (*n* = 1), and the effective electrode surface area (A) are also determined [65]. Hence, it was calculated that the I_pa_ of the bare GCE electrode was 33.87 µA and that the effective surface area was 6.61 × 10^−2^ cm^2^. After that, the active surface areas of the modified electrodes for the electrochemical immunosensor were found as depicted in Appendix A.

In addition, scans were performed at rates ranging from 10 to 70 mV/s in redox electrolyte for the TiO_2_-CS bio-nanocomposite as well as the electrochemical immunosensor to test the reversibility of the electrochemical immunosensor. The voltammetric profile that was acquired through CV was symmetrical, which means that the anodic peak current values increased as the scan rate increased, while the cathodic peak current values decreased as the scan rate increased. This suggests that the reaction was controlled by diffusion and that the electron transfer kinetics were reversible [66]. The scan rate acquired by CV was analyzed to develop regression curves of peak currents recorded in each cycle v/s square root of scan rate (*ν*^1/2^). It was revealed that the correlation between anodic (I_pa_) and cathodic peak currents (I_pc_) v/s *ν*^1/2^ was linear for both the TiO_2_-CS bio-nanocomposite and the constructed electrochemical immunosensor (Figure 4c). Excellent linearity has been shown in respective regression curves by the related computations, which is corroborated by the values of the regression coefficients (R^2^).
I_pa_ (GCE/TiO_2_-CS) (µA) = 0.112 *ν*^1/2^ + 9.336, R^2^ = 0.99825 (3)
I_pc_ (GCE/TiO_2_-CS) (µA) = −6.108 *ν*^1/2^ − 0.112, R^2^ = 0.99664(4)
I_pa_ (GCE/TiO_2_-CS/Antigen/BSA) (µA) = 0.157 *ν*^1/2^ + 6.975, R^2^ = 0.98909 (5)
I_pc_ (GCE/TiO_2_-CS/Antigen/BSA) (µA) = −4.134 *ν*^1/2^ − 0.104, R^2^ = 0.99925 (6)

An EIS analysis was carried out by utilizing an equivalent circuit to describe the characterization of a changed electrode/electrolyte junction after the deposition of various biomolecule layers. The impedance patterns that were obtained after each layer were modeled using the equivalent electrical circuit that is shown in the inset of the Nyquist plot (Figure 4d).

The resistance of the solution in series with the ohmic contacts makes up the circuit (R_s_). A resistance that is proportional to the charge transfer rate of the redox reaction taking place at the bio-functionalized electrode, abbreviated as R_ct_. A constant phase element (CPE) is connected to the capacitance of the interface between the biofunctionalized electrode and the electrolyte. Due to the nano-biofilm roughness and porosity, CPE reflects the non-ideal behavior of the double layer at the bio-functionalized electrode-electrolyte interface which will keep changing with the modification of the electrode. This is because the double layer is not ideal. The Warburg element is a special kind of electrochemical element that is involved in diffusion (Z_w_) [67].

Then, EIS data illustration was required in every step of the electrode surface alteration. Figure 4d shows a Nyquist plot with real and imaginary impedance components on the x and y axes, respectively. The acquired impedance data have a direct relationship with frequency. The semicircle section of the graph produced at high frequencies represents the capacity of electron transfer from the electrode surface to the ferri/ferrocyanide redox couple solution, also known as charge transfer resistance (R_ct_). At low-frequency values, the straight-line section of the curvature reveals diffusion-controlled nature at the planar electrode, showing reaction time. To evaluate the R_ct_ parameters for every surface-altered electrode, the following equation was used:(7)$$R_{\mathrm{ct}}=R_{p}-R_{s}$$
where R_p_ is the polarization resistance and R_s_ is the ferri/ferrocyanide redox couple solution resistance. The respective R_p_, R_ct_, and R_s_ parameters of each surface-altered electrode are reported in Appendix A.

According to the Nyquist plot, the blank GCE has a maximum R_ct_ value (biggest semicircle) in comparison with the TiO_2_ NPs-altered working electrode; the presence of TiO_2_ NPs facilitates electron transfer and improves current responsiveness by lowering the charge transfer resistance value, implying that TiO_2_ NPs might enable analyte adsorption and enrichment on the electrode surface [68,69]. The R_ct_ value of the TiO_2_-CS bio-nanocomposite is the minimum (smallest semicircle), indicating that the TiO_2_-CS bio-nanocomposite has a very high conducting nature (GCE/TiO_2_-CS). The R_ct_ value increased when the SARS-CoV-2 antigen was immobilized on the electrode surface, indicating that the proteins were successfully bound to the surface (GCE/TiO_2_-CS/Antigen). Furthermore, BSA occluding for boundless sites on the electrode, the R_ct_ value was high, indicating that, on the electrode surface (GCE/TiO_2_-CS/Antigen/BSA), the unbound sites were effectively blocked. Figure 4e,f includes bode graphs that illustrate the association between the frequency with the phase shift and amplitude, respectively. The phase v/s frequency bode plot (Figure 4f) depicts the maximum phase value recorded at a frequency value of ~1000 Hz. Therefore, the maximum change in capacitance effect was observed to reside within the specific frequency region. On the other hand, considering the Warburg element (Figure 4e) its vital effect at low frequencies is shown; we can relate this finding to the fact that the electron transfer process was limited by the mass transport of the electroactive species. The relation between frequency and impedance can be observed by the bode plot, which can also indicate how quickly the reaction takes place [70]. The relevant electrochemical data indicated that the electrochemical immunosensor was successfully fabricated. These EIS findings are congruent with the CV and DPV findings.

### 3.5. Electrochemical Detection

The functionality of the designed electrochemical immunosensors was evaluated using DPV to determine the SARS-CoV-2 antibody. Concentrations of SARS-CoV-2 antibody ranging from 50 ag mL^−1^ to 1 ng mL^−1^ were used to check the detection ability of the electrochemical immunosensor. In the absence of the antibody, the maximum current of 44.5 µA of the electrochemical immunosensor was observed at 0.20 V potential. The current retaliation was directly correlated with the concentration of the SARS-CoV-2 antibody, as shown in Figure 5a. A comparable drop in the current magnitude was seen when the antibody concentration raised from 50 ag mL^−1^ to 1 ng mL^−1^. The binding between the molecules of protein that forms a complex with the TiO_2_-CS bio-nanocomposite hinders the flow of electrons. The formation of the complex caused an electron transfer decrease as the antibody concentration in the redox electrolyte elevated [71]. The electrochemical immunosensor demonstrated good linearity throughout a linear range of 50 ag mL^−1^ to 1 ng mL^−1^, as shown in Figure 5b. The limit of detection (LOD) and limit of quantitation (LOQ) were computed using the standard equations LOQ = (10 × SD)/S and LOD = (3.3 × SD)/S, respectively. Here, SD is the standard deviation of the response of the calibration plot, SD is the standard error (SE) of intercept × N^1/2^, N is the number of samples, and S is the slope of the calibration plot. The LOD and LOQ were calculated to be 3.425 ag mL^−1^ and 10.381 ag mL^−1^, respectively. The current response from the calibration curve was calculated as follows:(8)$$\Delta I\;\left(\mathrm{immunosenor}\right)\left(µA\right)=1.07555\;\mathrm{Log}\;C\;(ag\;mL^{-1}+11.90897)$$
(9)$$R^{2}=0.97491$$

### 3.6. Stability and Selectivity Studies

The stability and selectivity of electrochemical immunosensors are significant parameters to analyze the performance of electrochemical immunosensor. The stability investigation was carried out by fabricating four electrochemical immunosensors on the same day and keeping them at 4 °C. Voltammetric detections of 50 ng mL^−1^ of the SARS-CoV-2 antibody were performed on different days over a range of 1–31 days, as shown in Figure 6a. The results show that there is no substantial change in the sensor response even after keeping it for 10 days, showing that the electrochemical immunosensor is stable. The electrochemical immunosensor response degrades while keeping it for one month. For the selectivity study, different interference analytes such as hemoglobin, transferrin, and dopamine compared with the SARS-CoV-2 antibody were detected with the same concentration, i.e., 50 ng mL^−1^, on the developed electrochemical immunosensor. The results are shown in Figure 6b evidence that the current response of interfering elements was substantially lower than that of the SARS-CoV-2 antibody. This suggests that the fabricated electrochemical immunosensor has high selectivity toward SARS-CoV-2 antibody detection.

## 4. Conclusions

In summary, we have developed a TiO_2_-CS bio-nanocomposite-based electrochemical immunosensor for the detection of the SARS-CoV-2 antibody. The TiO_2_-CS bio-nanocomposite was synthesized through a simple sol-gel method. The prepared TiO_2_-CS bio-nanocomposite possesses outstanding features such as high surface area, abundant surface functionalities, uniform film thickness, and high stability. The electrochemical studies reveal exemplary high electron mobility, redox behavior, and high binding affinity in the TiO_2_-CS bio-nanocomposite-modified electrodes. The electrochemical probe GCE/TiO_2_-CS/Antigen/BSA has been successfully fabricated, as evidenced by the electrochemical characterization. The fabricated electrochemical immunosensor exhibited excellent analytical performance with a LOD of 3.42 ag mL^−1^ in a wide linear range from 50 ag mL^−1^ to 1 ng mL^−1^. The electrochemical immunosensor platform exhibits superior stability and selectivity for detecting the SARS-CoV-2 antibody. The electrochemical immunosensing platform shows great potential to be effectively utilized in further real-sample diagnostics and high-throughput sensing of other viral infections.

## Data Availability

The data presented in this study are available in Appendix A.

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
