# Peer review of "Functionalized Titanium Dioxide Nanoparticle-Based Electrochemical Immunosensor for Detection of SARS-CoV-2 Antibody"

_diagnostics, 2022, doi:10.3390/diagnostics12112612_

Round 1

Reviewer 1 Report

To the Authors

The authors describe the development of a TiO2-CS bio-nanocomposite-based electrochemical immunosensor for the detection of the SARS-CoV-2 antibody

Fig. 4 needs to be reviewed to improve the resolution

Please, define the units ag mL-1 and ng mL-1. These units are not common for a very large audience.

When the authors say: “Concentrations of SARS-CoV-2 antibody ranging from 50 ag mL-1 to 1 ng mL-1 were used to check the detection ability of the electrochemical immunosensor.”, it did not become clear why they used these values. What is the concentration of SARS-CoV-2 antibody found in the blood of a patient that presents symptoms of SARS-CoV-2?

Reviewer 2 Report

In their manuscript “Functionalized Titanium Dioxide Nanoparticles based Electrochemical Immunosensor for Detection of SARS-CoV-2 Antibody” the authors describe fabrication and testing of an electrochemical immunosensor for SARS-Cov2 antibody detection. They first assembled a TiO2-Chitosan biocomposite, functionalized a glassy carbon electrode (GCE) with the material, and finally coupled via EDC/NHS chemistry the nuclear protein of SARS-CoV2 as antigen. After passivation of the GCE with BSA antibody binding studies were performed.  

The introduction of the study gives a good review of current research in the field of nanocomposite based electrochemical detection methods. The fabrication methods are well described and easy to follow, ditto for the extensive material characterization steps. The results are conclusive.

Nevertheless, some points need a clarification. The assays does not detect the antigen rather the immunological response of the adaptive immune system, specific antibodies. Is this a true medical need in the field of diagnostics?

In the real world the antibodies are detected from serum samples. Did you check this with solutions of spiked antibodies? Complex media might be a problem for selective and sensitive detection.

In Table 1 you don´t detect the virus as indicated in the heading.

Minor points: 

Please check hyphenation in the manuscript!

Please check spelling (spaces) throughout the manuscript of measures: X%, Y °C, Z mg

Check consistent spelling: UV-vis, UV-vis

Change: mediums to media

Page 2: "… allowing them to produce spontaneous findings". I don´t get the sense!

Change: easy-to use to: ease of use  

Introduce abbreviation: S-RDB

Page 3, insert verb: The electrode was bath sonicated with 0.1M of H2SO4 for 5 minutes and then washed with ethanol and water alternatively, for 2 minutes each.

Correct: 2.5. Assembly of  the Immunosensing Platform for SARS-CoV-2 antibody detection

Page 5, incomplete sentence, verb missing: Hence the successful synthesis of TiO2-CS bio-nanocomposite.

Page 7: delete in: It seems that the TiO2-CS bio-nanocomposite in (Figure 3 f) is semicrystalline,
